# Growth of Zr/ZrO₂ Core–Shell Structures by Fast Thermal Oxidation

**Juan Francisco Ramos-Justicia *** [ID]**, José Luis Ballester-Andújar, Ana Urbieta** [ID] **and Paloma Fernández** [ID]

Department Materials Physics, Faculty of Physics, University Complutense, Plaza de Ciencias, 1,
28040 Madrid, Spain
* Correspondence: juanfrra@ucm.es

**Abstract:** This research has been conducted to characterize and validate resistive heating as a synthesis method for zirconium oxides ($ZrO_2$). A wire of Zr has been oxidized to form a core–shell structure, in which the core is a metal wire, and the shell is an oxide layer that is around 10 μm thick. The characterization of the samples has been performed by means of several techniques based on Scanning Electron Microscopy (SEM). The topography images show that thermal gradient appears to have little influence on morphology, unlike time, which plays an important role. The chemical composition was analyzed by X-ray spectroscopy (EDX) and X-ray diffraction (XRD), and Raman spectroscopy has been used to assess crystallinity and crystal structure. The oxide layer is mainly formed by monoclinic $ZrO_2$, alongside other, less significant, phases. Photoluminescence (PL) and cathodoluminescence (CL) measurements have allowed us to study the distribution of defects along the shell and to confirm the degree of uniformity. The oxygen vacancies, either as isolated defects or forming complexes with impurities, play a determinant role in the luminescent processes. Color centers, mainly electron centers such as F, $F_A$ and $F_{AA}$, give rise to several visible emissions extending from blue to green, with main components at around 2 eV, 2.4–2.5 eV and 2.7 eV. The differences between PL and CL in relation to distinct recombination paths are also discussed.

**Keywords:** transition metal oxides; defects; luminescence; fast growth

## 1. Introduction

Transition metal oxides (TMO) are gaining relevance every day due to their wide range of applications, versatility and advantageous physical properties (conductive, magnetic, luminescent, etc.), enabling the fabrication of multifunctional systems. However, to make these applications viable in actuality, much deeper knowledge about critical factors such as carrier concentration, recombination rates, mobility, defect structure and their influence on the band structure (and hence, on most relevant properties), is needed [1]. Uses in sectors as different as the Internet of Things (IoT), health or green energy demand high-performance materials; sensors capable of detecting the presence of toxic gases or monitoring changes in critical parameters for human health may be used to trigger corrective or warning protocols [2]; more efficient catalysts might help to improve water cleaning or to ease hydrogen production processes [3], which are crucial for attaining a cleaner environment and a safer, fairer and more equal society.

Some of these applications have a common bottleneck: the low efficiency of the redox reactions involved. The prerequisite for a TMO to be efficient as a redox mediator is that the redox potential for the radicals or reactants entailed lies within the band gap of the oxide (for instance, in the case of photocatalysis, $E_o$ ($H_2O$/•OH) = 2.8 V and $E_o$ ($O_2/O_2^{•−}$) = 0.28 V vs. NHE [4]). In this context, the study of the defect structure and its influence on the band structure of the materials utilized become of the utmost importance. Moreover, a more extensive study on the effect of impurities and defects would pave the way to improve their tailoring capabilities. Under this framework, the role of composite materials

is gaining ever more relevance since it opens the possibility to tune the band structure over a broader range, playing not only with the combination of materials, but designing new architectures optimized for different working conditions. The feasible combinations of materials that can be made are numerous, but the "active" part will always be the semiconductor. Some common configurations are core–shell structures, heterostructures and arrays of heterostructures [5,6].

For a long time, zirconium oxide has been a known TMO, present in more than 40 minerals, mainly oxides and silicates. The most abundant zirconium minerals are zircon ($ZrSiO_4$), baddeleyite ($ZrO_2$) and eudialyte ($(Na, Ca)_5 (Zr, Fe, Mn)[O, OH, Cl][Si_6O_{17}]$); the main source of this oxide is the baddeleyite mineral. Traditionally, the main uses of zirconium oxide have been those related to its corrosion resistance, refractory properties and toughness. However, new applications have arisen, and several review papers have been recently published [7–9]. Among the new roles found for $ZrO_2$, particularly at the micro-/nanoscale, those for environmental remediation and biomedical applications are gaining relevance [9]. Antibacterial and antifungal activities play a central role among biomedical applications, inasmuch as the same mechanisms underlie both of them. The first procedure proposed is an electrostatic interaction between the negatively charged bacteria membranes and the $ZrO_2$ particles. The high surface area would favor this interaction and lead to the inhibition of the key metabolic functions carried out by reactive oxygen species such as oxygen and hydroxyl super radicals ($\cdot O_2^-$, $\cdot OH$) [10,11]. A similar mechanism would operate in the case of antifungal activity, inhibiting cell division [12].

On the other hand, environmental remediation is probably the most pressing challenge faced by the present society [13]. The removal of antibiotics, textile dyes and emerging pollutants such as fluoride is becoming a serious problem. The review paper published by Van Tran et al. [9] contains a good revision of recent works on the use of zirconium oxide to eliminate different contaminants, with the added value that $ZrO_2$ is synthesized by different green routes, such as plant tissues, or microorganisms, such as bacteria, algae or fungi.

Besides these health or environmental applications, the use of zirconium oxide is increasing in importance in several optoelectronic devices and systems requiring higher permittivities, such as energy storage devices or Resistive Random Access Memories [14]. An important goal for optoelectronic applications is to optimize the doping processes. There is a general agreement on the relevant role of defects and dopants on the optoelectronic properties [9,15–18]. Dopants do not only interact with intrinsic defects forming different complexes, but they can also dramatically modify the band structure of the material. $ZrO_2$ is a very wide bandgap material (>5 eV), and consequently, all processes involving photoexcitation require ultraviolet illumination. The introduction of a dopant such as Ni causes a narrowing of the bandgap below 2.7 eV [19], thus enabling its use in micro-/nanoelectronic devices.

In the present work we report on the fabrication of core–shell structures ($Zr/ZrO_2$) by thermal oxidation of the metal. Thermal oxidation occurs when an electrical current passes through a metal, this is called the Joule effect, while the metal is kept in an ambient atmosphere. This procedure has already been proven to be effective in different oxides such as ZnO or CuO, being also suitable for the growth of doped oxide shells [6,20]. However, fewer studies have been performed on other transition metal oxides such as $ZrO_2$. This method is simple and fast, introducing multiple possibilities for the fabrication of doped oxides and electronic composites. A detailed study of the growth conditions, morphology, crystalline structure and composition of the oxide layer has been carried out. In addition, its luminescence properties have been investigated, combining cathodo- and photoluminescence experiments to analyze the defect structure of the material.

## 2. Experimental Method

The experimental setup used to grow the samples is shown in Figure 1a. A wire made of metal that is to be oxidized is suspended between two electrodes connected to a current

source. The full system was kept in an ambient atmosphere. As a consequence of Joule heating, a temperature gradient was established along the wire, promoting the oxidation of the metal and yielding core–shell structures, in which the core is the native metal. In order to determine the thickness of the oxide layer, which was around 10 μm, we took the average of some radii of the circular crown formed by the oxide layer.

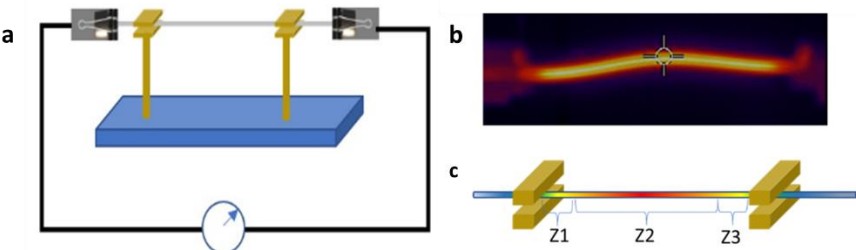

**Figure 1.** (**a**) Experimental setup for resistive heating growth. (**b**) Thermographic image of a wire. The cross indicates the point at which the maximum temperature is measured. (**c**) Sketch of the three zones established for the characterization of the core–shell structures grown.

To study the influence of the growth parameters (current and time), different sets of samples were prepared.

The starting material is a commercial Zr wire (Goodfellows ZR00-WR-000130), as drawn, with a nominal purity of 99.2%. According to the provider's specifications, the main impurity content corresponds to hafnium (2500 ppm). Other transition metals such as chromium or iron were also present but at much lower concentrations (200 ppm). The wire length (distance between electrodes) is 10 cm, and the diameter is 0.25 mm.

According to the temperature profile established in the wire, three different zones are defined for characterization purposes. Figure 1 shows a thermal image of the wire (b) and the zones defined (c). Thermal images were recorded using an FLIR E8-XT thermographic camera. The maximum temperatures reached were between 430 and 500 °C. Temperatures were measured using an optical pyrometer Infratherm IGA 12-S. There are some analytical models of the stationary distribution of temperature of the wire. A detailed explanation of the derivation of the differential equation that governs the temperature profile because of Joule heating can be found in [21], whose notation is kept below:

$$\kappa A_{cr}T''(x) - \left(h\pi d - J^2 A_{cr}\rho_0\xi\right)T(x) = J^2 A_{cr}\rho_0(\xi T_0 - 1) - h\pi dT_0$$

The general behavior of the temperature profile can be written in terms of real exponentials (hyperbolic functions) or complex exponentials (sines and cosines), independent of the initial conditions. Furthermore, this behavior depends solely on the sign of the zeroth derivative temperature term. The sign of this term can be positive or negative at will by modifying physical parameters such as intensity, diameter and resistivity (44 μΩcm at 20 °C for zirconium). If this term vanished, a parabolic profile would be observed. In our case, the model does not adjust for experimental data, even including corrective terms, such as radiative heat loss in the prior equation. We believe that there might be size factors related to the wire diameter (0.25 mm) or physical processes that should not be disregarded in a first approach, which would influence the accuracy of the model. A complementary, but extremely complex, model should account for the time evolution of the oxide shell. Nevertheless, this is out of our scope, since it would require detailed knowledge of the growth kinetics, the defect structure at each growth step and their influence on parameters such as the resistivity and thermal properties of zirconium oxide.

In this work, we have prepared two different sets of samples. The first set consists of five wires subjected to currents from 2.2 to 2.8 A for 10 s.

For the second set of samples, the maximum intensity was 2.2 A in all cases, but the current–time profiles and the times during which the wires were held at the maximum current were varied. Figure 2 sketches the current–time profile used.

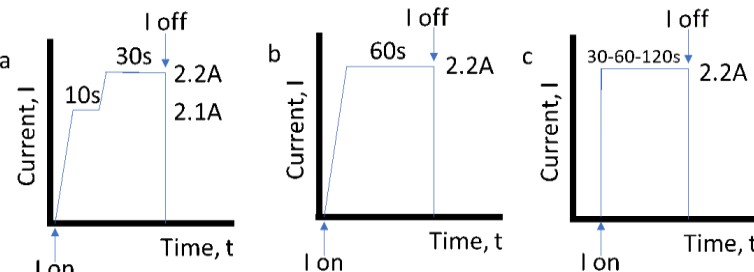

**Figure 2.** Current–time profiles used in the different experiments: (**a**) intensity gradually increased from 0 to 2.1 A and held 10 s, then risen slowly to 2.2 A and held 30 s; (**b**) intensity gradually increased from 0 to 2.2 A and afterwards held 60 s; (**c**) intensity directly established at 2.2 A and held 30, 60 and 120 s.

The characterization was carried out by means of SEM-based techniques and optical spectroscopy. Emissive mode images were recorded with an FEI Inspect SEM. X-ray microanalysis (EDX) was performed with an Hitachi TM 3000 SEM equipped with a Brucker AXS Quantax system. μ-photoluminescence (μ-PL) and Raman spectroscopy measurements were obtained using a confocal microscope Horiba Jobin-Ybon LabRAM HR800 with an excitation He-Cd laser source operated at 325 nm. The crystal structure was studied by X-ray diffraction using a PANalytical Empyrean with the Cu-K$_\alpha$ line.

## 3. Results and Discussion

X-ray diffraction patterns were recorded under a grazing angle to avoid the signal of the Zr metal core. However, due to the morphology and low thicknesses of most of the oxide shells, the patterns are very noisy. For that reason, in Figure 3, we show the comparison between the main peaks of the pattern and the peak lists of the corresponding phases according to JCDPS cards.

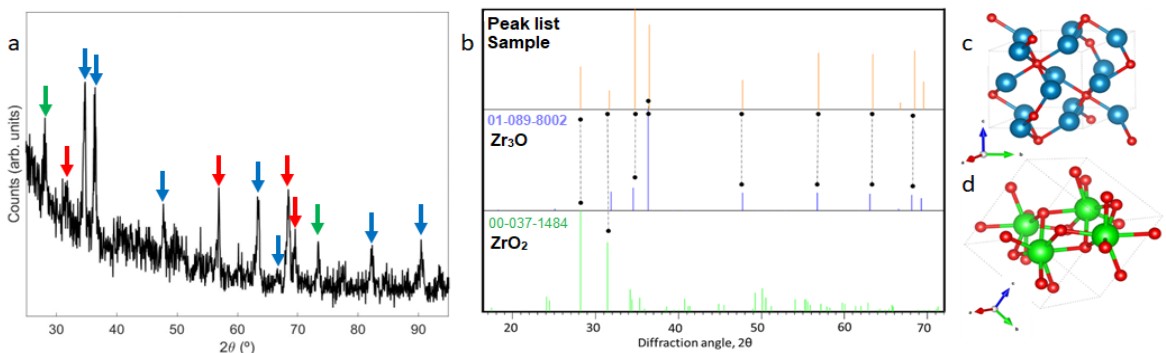

**Figure 3.** (**a**) XRD diffraction pattern; arrows mark the different phases found (green ones are for the monoclinic $ZrO_2$ phase, blue ones are for the hexagonal $Zr_3O$ phase and red ones are for peaks related to both phases). (**b**) Comparison of the peaks obtained (upper) to those listed in the JCPDS cards for the corresponding phases: $Zr_3O$ (middle) and $ZrO_2$ (lower). (**c**) Crystal structure of the $Zr_3O$ phase (O: red; Zr: blue). (**d**) Crystal structure of the $ZrO_2$ phase (O: red; Zr: green).

The XRD pattern is shown in Figure 3a. Figure 3b (upper part) shows the peak list obtained from a typical X-ray diffraction pattern from our samples. The peaks fit with the presence of two zirconium oxide phases. The peaks shown in the middle graph correspond to a non-stoichiometric $Zr_3O$ hexagonal phase (JCDPS 01-089-8002) with lattice

parameters a = 5.6172 Å, b = 5.6172 Å and c = 5.1850 Å (space group P63 2 2) (Figure 3c). The $ZrO_2$ monoclinic phase was identified with the JCDPS 00-037-1484 card, with lattice parameters a = 5.1473 Å, b = 5.2088 Å and c = 5.3166 Å (space group P21/c) (Figure 3d). No additional phases related to the presence of impurities were detected. The presence of non-stoichiometric phases might be expected since the oxidation process is extremely fast and might not be completed uniformly. On the other hand, this would lead to low-crystalline-quality areas that would be responsible for the noisy patterns.

To further assess the phase content, we performed μ-Raman experiments. Then, we recorded Raman spectra from local regions, particularly from the central parts of the wires where the shells are better formed, and the oxidation process was completed. The Raman spectra from all the samples studied are quite similar. An example of these typical Raman spectrum is shown in Figure 4. The peaks observed correspond to the different $A_g$ (102, 188, 308, 346, 475, 561 and 637 $cm^{-1}$) and $B_g$ modes (175, 221, 334, 381 and 614 $cm^{-1}$) of the monoclinic phase of $ZrO_2$. The peaks are labelled according to the data found in the works of Quintard et al. [22], Ishigame et al. [23] and Kumari et al. [24]. All the peaks reported previously for the monoclinic phase are observed, and most of them are well-defined and not very broad, so the oxide shell is meant to have a good crystal quality. Red labels correspond to O-O bonds; the green ones correspond to Zr-Zr bonds; the orange ones correspond to Zr-O bonds. The peak at 175 $cm^{-1}$ is assigned to a combination of both $A_g$ and $B_g$ modes. Some modes (for instance, that at 636 $cm^{-1}$) could be also ascribed to the tetragonal phase. Nevertheless, since only the monoclinic phase was revealed through the X-ray diffraction experiments, we discard this possibility, and we opt for assigning all the modes to the monoclinic structure.

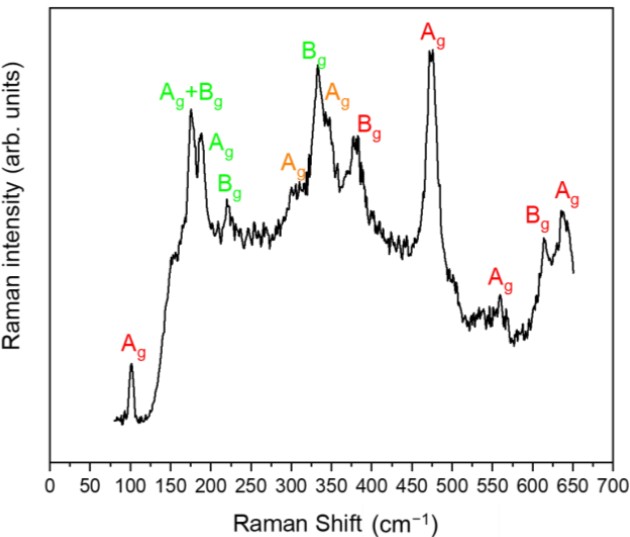

**Figure 4.** Typical Raman spectrum obtained from the samples. Colored labels: red, O-O bonds; green, Zr-Zr bonds; orange, Zr-O bonds.

The first set of samples described in the Experimental Method Section was used to check the influence of the temperature gradient established along the wire. Figure 5 shows typical topography images obtained from these samples, which upon inspection, do not appear to show any big differences between them. At low magnification (Figure 5a,b) a form of columnar growth is observed: this morphology reflects the drawing tracks from the metal wire. This pattern is similar in all the samples, although in those grown at higher intensities (b), the oxide layer is thicker, and therefore, the grain structure is better defined. Part c of this figure shows a detail of the oxide layer, where the formed grains are clearly visible. The charge effects observed in this image also hint to the thickness of the oxide layer.

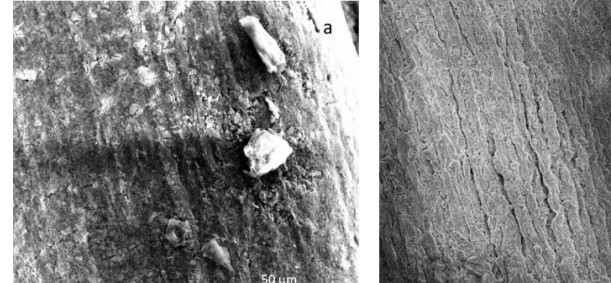
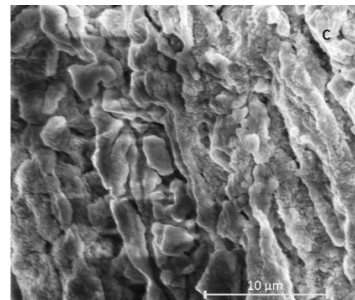

**Figure 5.** SEM images obtained from simples grown at a current of: (**a**) 2.2 A; (**b**) 2.8 A; (**c**) detail of the sample shown in part (**b**).

In the second set of samples, we focused on the growth of microstructures and their evolution with treatment time. A current of 2.2 A was fixed in all cases; the difference resides in the time profile. In the samples grown for the shortest times, (time profiles shown in Figure 2a–c for 30 s), only the first stages of microstructure growth are observed. As shown in Figure 6, small needles grew between the grooves associated to the drawing lines.

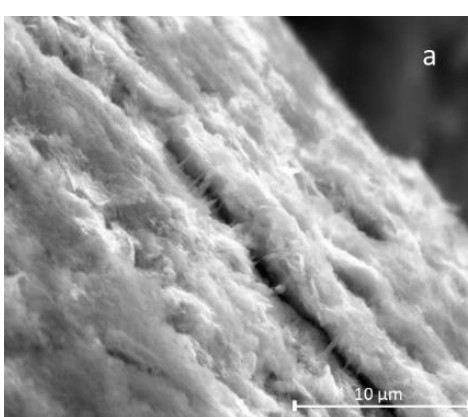
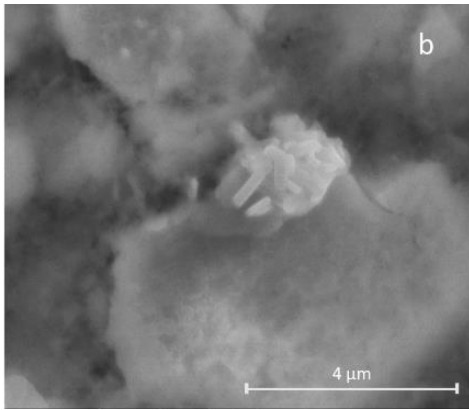

**Figure 6.** (**a**,**b**): SEM images of the nanostructures obtained at a current of 2.2 A for 30 s which shows needles between the grooves associated to the drawing lines.

The X-ray microanalysis shows a homogeneous distribution of Zr and O across the shell. Figure 7a shows a cross-section of the wire. Figure 7b,c show the maps and line profiles corresponding to O and Zr, respectively. The maps evidence that oxygen was only detected from the outer part corresponding to the oxidized layer. As aforementioned, the thickness of the shell (about 10 μm) was taken as the average of some of the circular crown radii of the oxide layer. The profile line shown in Figure 8c strengthens the prior estimation. The spectrum shown in Figure 7d was recorded only from the shell (not made in cross-section) and must be carefully considered. The spectra were recorded at an accelerating voltage of 10 keV to minimize the contribution of the zirconium core to the signal. However, the $K_\alpha$ line from Zr (15.74 keV) is not perceptible in these conditions, and the characteristic line used for Zr is $L_\alpha$, with an energy of 2.04 keV, albeit its intensity was 25 times lower [25]. This estimation is based on the ratio of average intensity fluorescence coefficients of $K_\alpha$ and $L_\alpha$ shells [25]. That is the reason why the most intense signals appear to come from Si and S, whose $K_\alpha$ lines are placed at 1.739 and 2.307 keV, respectively. According to the provider, the main impurities are Hf, O and C. Nevertheless, as shown in Figure 7d, Si and S can also be observed, and these were possibly added to Zr in the purification process from mineral ore. The carbon peak is due to graphite tape, which was used to stick the sample to the holder. The inset of Figure 7d shows an estimation of the relative intensities of both Zr lines.

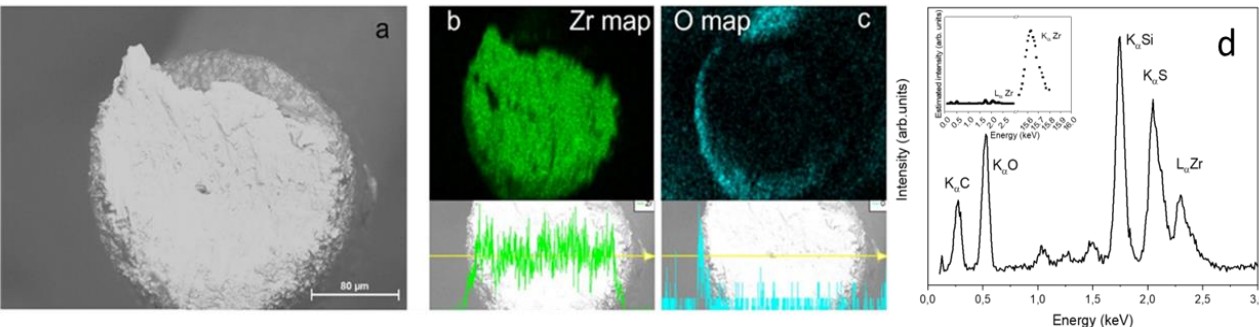

**Figure 7.** (**a**) Cross-section of the wire; (**b**) Zr map (upper) and profile (lower), showing the uniformity of Zr distribution; (**c**) O map (upper) and profile (lower), the oxygen signal is significant only at the outer surface of the wire (the lack of O signal at the right border is a shadow effect due to the topography of the sample and its orientation respect to the detector); (**d**) EDX spectrum recorded with an accelerating voltage of 10 keV on the surface of the wire. The inset shows an estimation of the intensity of the Zr $K_\alpha$ line relative to the peaks shown in the main graph.

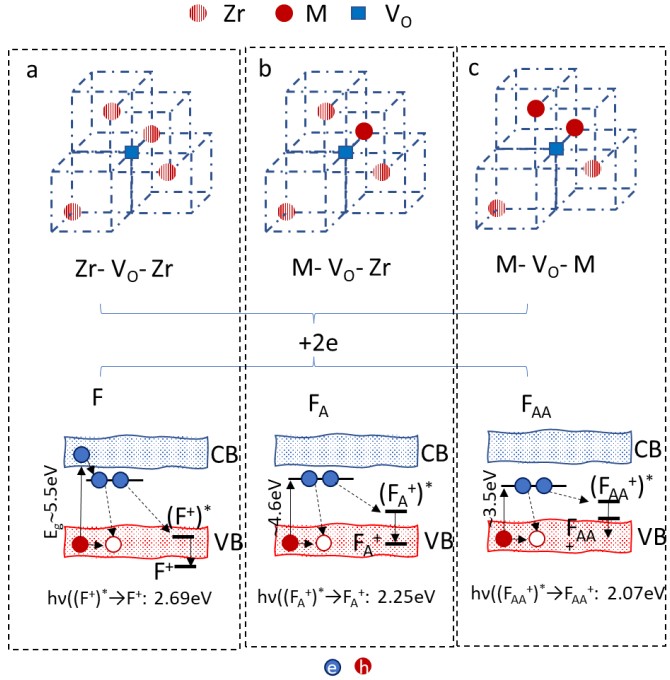

**Figure 8.** Schematic representation of the different F centers expected in this system [9–12] (upper) and the corresponding recombination paths (lower). (**a**) Reactions 1, 3 and 4; (**b**) reactions 5 and 6; (**c**) reactions 7 and 8. The asterisk (*) means excited state.

Luminescence properties were studied by means of photoluminescence (PL) and cathodoluminescence (CL). Depending on the excitation mechanism (PL/CL), different routes for valence electrons promoted to the conduction band could be enhanced, so complementary information about the recombination paths and defect structure can be obtained using both techniques. In both cases, the spectra consist of a broad visible band extending from blue to green, although the relative intensity of the different components depends strongly on the excitation source. The origin of these emissions is not clear [26], but there is a consensus view about the role of defects and residual impurities on them [17,27,28]. The majority of the defects found on oxides are color centers. The most common ones are F- or $F_2$-type centers, oxygen vacancies or divacancies with trapped electrons, respectively. However, T-centers composed of two oxygen vacancies and a Zr cation ($V_O$-Zr-$V_O$), hole centers such as oxygen with a trapped hole ($O^-$) or V-type defects are also present [29]. The

mechanisms behind the luminescence in $ZrO_2$ are not well understood yet. Even so, most of the previous works rely on the relation of the visible emission with oxygen vacancies and complexes $V_O$-M, where M is a cationic impurity that would occupy Zr sites [18,27,28] (see the references therein). In this case, we refer to F, $F_A$ and $F_{AA}$ centers as the main ones responsible for the visible emission bands.

Figure 8 shows a sketch of the three types of F centers present in the material. The F center (part a) is formed when an oxygen vacancy adjacent to two Zr ions traps two electrons [Zr- $V_O$- Zr]. The centers shown in parts b and c are extrinsic since they involve the cationic impurity. In the first case, (Figure 8b) Zr and the impurity cations are the nearest neighbors of the oxygen vacancy [M- $V_O$- Zr]. Finally, part c shows the defect formed when the two nearest neighbors are cationic impurities [M- $V_O$- M]. The most frequently reported impurities are $Ti^{+4}$ (or $Ti^{+3}$), as residual impurities that cannot be eliminated, and $Y^{+3}$, since yttria is one of the compounds used to stabilize the zirconia. According to the specifications of the provider, the main impurity present in the Zr wire is hafnium, which is also mentioned as a candidate for cationic impurity [30]. The defect reactions that would give rise to the visible emissions according to [31] are written below. The first step is the excitation of electrons through the band gap

$$\mathrm{ZrO_2} + h\nu \ (5.82 \ \mathrm{eV}) \rightarrow e^- + h^+ \tag{1}$$

Once the electron–hole pairs are created, several recombination paths could be activated. In our system, the electrons promoted to the conduction band could be easily trapped by the singly ionized oxygen vacancies ($F^+$ centers)

$$F^+ + e^- \rightarrow F \tag{2}$$

The trapping of a hole by the F center would create an excited $F^+$ level

$$F + h^+ \rightarrow \left(F^+\right)^* \tag{3}$$

When the ground state is recovered

$$\left(F^+\right)^* \rightarrow F^+ + h\nu \ (2.69 \ \mathrm{eV}) \tag{4}$$

The defect reactions for the extrinsic $F$ centers are written as [18,31,32]

$$F_A^+ + h\nu \ (4.66 \ \mathrm{eV}) \rightarrow F_A + h^+ \tag{5}$$

$$F_A + h^+ \rightarrow \left(F_A^+\right)^* \rightarrow F_A^+ + h\nu \ (2.25 \ \mathrm{eV}) \tag{6}$$

$$F_{AA}^+ + h\nu \ (3.49 \ \mathrm{eV}) \rightarrow F_{AA} + h^+ \tag{7}$$

$$F_{AA} + h^+ \rightarrow \left(F_{AA}^+\right)^* \rightarrow F_{AA}^+ + h\nu \ (2.07 \ \mathrm{eV}) \tag{8}$$

The typical photoluminescence (PL) spectrum recorded under excitation with a 325 nm He-Cd laser (Figure 9a) exhibits a broad band centered around 2.4 eV. A similar band that peaked around 2.4–2.5 eV has been previously reported [33–35], and it is attributed to oxygen-vacancy-related centers.

The deconvolution of this band in three components fits quite well with the transitions involving the F centers mentioned above [32,36]. It is worthy to note that the position of the electronic levels associated with defects are very sensitive to the surroundings and lattice relaxation states, which could play an important role in our case due to the low oxygen stoichiometry. The relative weight of the different components reflects the fact that the excitation source has an energy of 3.8 eV (325 nm), and therefore electrons cannot be directly promoted to the conduction band, and F and $(F^+)^*$ must be filled directly with electrons from the valence band or different defect levels with less energy. In this case, with an excitation value of 3.8 eV, the transitions involving $F_A^+$ and $(F_A^+)^*$ seem to be more

favored, and hence, the green emission predominates. Previous works [15–18,30,32,36] report a broad blue band centered at 2.7 eV, but either the excitation sources are high-energy photons (above the bandgap) [15,16,31,32,34,35,37] or they refer to cathodoluminescence experiments [18,30,33,36]. The cathodoluminescence experiments performed in this work also point in this direction (Figure 9b). In this case, the spectrum is centered closer to 2.6 eV, and the band is narrower than it is in the PL case. The deconvolution of the visible CL band shows three components at positions such as those obtained in PL, but with different relative intensities. In fact, the more intense component is now the blue one (2.7 eV). In cathodoluminescence, if we excite the emission with high-energy electrons (15–20 keV), then the valence band electrons may be easily promoted to the conduction band, so new and more efficient recombination paths could be favored. The model proposed by Wang et al. [36] attributes the 2.5 eV emission to a transition involving the cationic impurity (Ti in the work of Wang). Electrons trapped by a shallow donor ($V_O^{\bullet}$) could be released either to the conduction band or to the F centers. If they are released to the CB, they could recombine with holes in the valence band, but also through the levels introduced by the oxygen vacancy–cationic impurity complexes, as depicted in the scheme in Figure 10 adapted from [36].

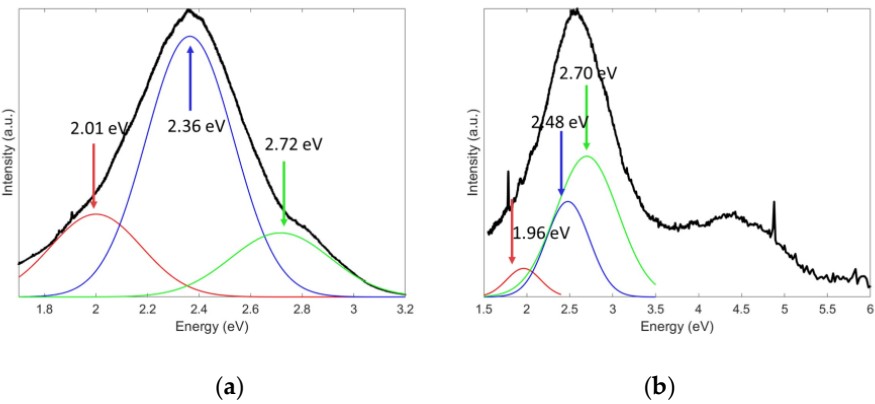

(**a**)                                        (**b**)

**Figure 9.** Characteristic luminescence spectra: (**a**) PL at an excitation wavelength of 325 nm; (**b**) CL at an accelerating voltage of 20 keV.

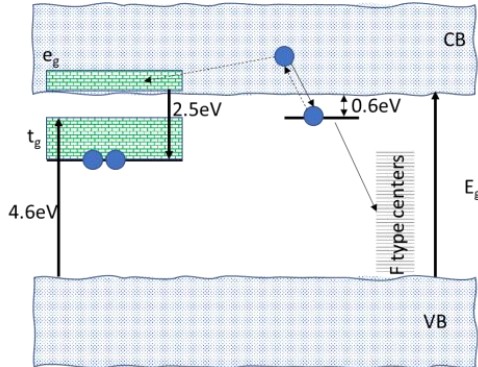

**Figure 10.** Recombination paths in CL as proposed by Wang et al. [36].

Finally, the band observed in the CL spectrum centered at 4.5 eV (Figure 9b) could also be associated with these kind of complex centers [36].

## 4. Conclusions

Zr/ZrO$_2$ core–shell structures were obtained by the rapid thermal oxidation of Zr wire. The highest quality of the layer was achieved at the center of the wire, where temperatures of 430 to 500 °C were reached. The shell obtained seemed to be quite uniform, except

at the extremes, where the temperature reached was lower. The thickness of the layer was calculated by averaging several radii of the circular crown and was around 10 μm thick. The oxygen profile line supports this result. The zirconium oxide constituting the shell has a monoclinic structure, although due to the lack of stoichiometry, a $Zr_3O$ phase was also detected. This phase appears to have no major influence on the luminescent properties or crystal quality. Raman spectroscopy revealed the characteristic peaks of the monoclinic phase without relevant broadening or shifting with respect to the reported values. Luminescence spectra show a broad visible band that can be deconvoluted into three components at 2 eV, 2.4–2.5 eV and 2.7 eV. All the emissions can be attributed to electron centers. Different kinds of F centers (oxygen vacancy related) let us build a consistent scenario to explain the luminescent behavior. The deconvolution peaks of the visible band of cathodoluminescence spectra are nearly the same as those obtained in the PL spectra, although their relative intensities are different.

**Author Contributions:** Conceptualization, A.U. and P.F.; Formal analysis, J.F.R.-J., J.L.B.-A., A.U. and P.F.; Funding acquisition, P.F.; Investigation, J.F.R.-J., J.L.B.-A. and A.U.; Methodology, A.U. and P.F.; Project administration, P.F.; Resources, P.F.; Supervision, A.U. and P.F.; Validation, J.F.R.-J., J.L.B.-A., A.U. and P.F.; Visualization, J.F.R.-J., J.L.B.-A., A.U. and P.F.; Writing—original draft, J.F.R.-J., J.L.B.-A. and P.F.; Writing—review and editing, A.U. and P.F. All authors have read and agreed to the published version of the manuscript.

**Funding:** This work has been funded by the Complutense University—Comunidad de Madrid via project PR65/19-22464.

**Data Availability Statement:** The data presented in this study are available on request from the corresponding author. The data are not publicly available due to technical reasons.

**Conflicts of Interest:** The authors declare no conflict of interest.

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
