# Peer review of "Growth of Zr/ZrO2 Core–Shell Structures by Fast Thermal Oxidation"

_applsci, doi:10.3390/app13063714_

Round 1

Reviewer 1 Report

Dear Authors,

After a detailed analysis of your submission, I would like to inform you that I recommend your paper for publication but with some major changes, which needed to be done before.

The research paper with the title “Growth of Zr/ZrO2 core-shell structures by Fast Thermal Oxidation” has a reasonable arrangement and the presented results are interesting from the point of view of material science.  In my opinion, the experiment is planned correctly but the discussion suffers because of the lack of some important information.

My annotations are as follows:

1)      In line 99, you claim that the zirconium oxide layer is around 10um thick but anywhere in the article is information on how it was measured/estimated. Please, explain how it was estimated and add detailed discussion in the text.

2)      XRD analysis is done very poorly. There is a lack of an original pattern, which should be shown in the article regardless of the noisy character. You extracted some prominent peaks from it, but in this way, you can not even roughly estimate which type of structure is in the material. Detailed analysis of the original diffractogram should be done i.e phase analysis, estimation of the lattice parameters, etc.  In my opinion, due to differences in the temperature distribution along the wire, there should be at least three diffractograms done in the different spots of the sample. Moreover, there is no clear information on which type of structure has the investigated material. Thorough article I found at least 3 (tetragonal, hexagonal, and monoclinic), and the discussion in some parts of your paper is not concise with your XRD results as it is e.g in the part about Raman. Also in conclusion is not clear what structure is in the investigated material.

3)      How you estimated the thickness of zirconia oxide from the SEM results? I was only able to spot in the X-ray microanalysis that the oxide layer is not uniform.

4)      As far as I understand, PL and CL measurements were done for the purpose of detecting defects in the structure and explanation of their role in the luminescent processes. But still, I know nothing about it, and how it is related to structural investigations.

5)      Going through the text I’ve found at least two parts, where some phrases are duplicated. See lines 9-13 and 71-75.

6)      In my opinion figs 1-3 should be merged into one figure due to the readability of the text. The same story with Figs 6-7, and 8-9. All figures should be placed in the text directly after the part where you mention them. I was quite surprised when I saw the last fig. after the conclusions.

7)      Fig. 5 is not mentioned in the text

Author Response

Dear Editor

Concerning the manuscript referenced applsci-2237059, and entitled “Growth of Zr/ZrO2 core-shell structures by Fast Thermal Oxidation”, we have revised the manuscript according to the reviewers’ comments. We thank their carefully work  what, doubtlessly, will improve the quality of the paper. Find below the list of changes made in the manuscript, where they have been marked in yellow.

Reviewer #1

Dear Authors,

After a detailed analysis of your submission, I would like to inform you that I recommend your paper for publication but with some major changes, which needed to be done before.

The research paper with the title “Growth of Zr/ZrO2 core-shell structures by Fast Thermal Oxidation” has a reasonable arrangement and the presented results are interesting from the point of view of material science.  In my opinion, the experiment is planned correctly but the discussion suffers because of the lack of some important information.

My annotations are as follows:

1)      In line 99, you claim that the zirconium oxide layer is around 10um thick but anywhere in the article is information on how it was measured/estimated. Please, explain how it was estimated and add detailed discussion in the text.

A sentence has been included at this point to clarify how we have measured the thickness of the oxide layer.

2)      XRD analysis is done very poorly. There is a lack of an original pattern, which should be shown in the article regardless of the noisy character. You extracted some prominent peaks from it, but in this way, you can not even roughly estimate which type of structure is in the material. Detailed analysis of the original diffractogram should be done i.e phase analysis, estimation of the lattice parameters, etc.  In my opinion, due to differences in the temperature distribution along the wire, there should be at least three diffractograms done in the different spots of the sample. Moreover, there is no clear information on which type of structure has the investigated material. Thorough article I found at least 3 (tetragonal, hexagonal, and monoclinic), and the discussion in some parts of your paper is not concise with your XRD results as it is e.g in the part about Raman. Also in conclusion is not clear what structure is in the investigated material.

As mentioned in the manuscript, XRD patterns are noisy, as shown in the graph below. We have performed the phase analysis and included it in the manuscript as figure 3. However, a more detailed study involving lattice parameters seems to be not accurate.

In fact, there are only two different regions according to temperature profile, i.e. the center of the wire and both ends. We have performed XRD in both regions and found no significant differences in the patterns.

Regarding the different structures (monoclinic, tetragonal or hexagonal) the reviewer comment, some parts of the text and the abstract have been modified to point out that the monoclinic structure predominates in the oxide layer. However, some XRD peaks related to a non-stoichiometric hexagonal phase have been observed and some of the Raman modes detected may be associated to both tetragonal and monoclinic phases.

3)      How you estimated the thickness of zirconia oxide from the SEM results? I was only able to spot in the X-ray microanalysis that the oxide layer is not uniform.

We have included a sentence in the experimental section as well as in the results section to explain this point.

4)      As far as I understand, PL and CL measurements were done for the purpose of detecting defects in the structure and explanation of their role in the luminescent processes. But still, I know nothing about it, and how it is related to structural investigations.

We do not understand the referee comment. Nevertheless, a sentence explaining the differences between CL and PL has been included to clarify the convenience of using both.

5)      Going through the text I’ve found at least two parts, where some phrases are duplicated. See lines 9-13 and 71-75.

All the text has been revised and duplications have been delated.

6)      In my opinion figs 1-3 should be merged into one figure due to the readability of the text. The same story with Figs 6-7, and 8-9. All figures should be placed in the text directly after the part where you mention them. I was quite surprised when I saw the last fig. after the conclusions.

We thank the reviewer for this appreciation.

We agree that figures 1 and 2 can be merged (to current figure 1) since they are referring to the same process, however, in our opinion it is better to maintain figure 3 (current figure 2) separated.

Regarding figures 6 and 7 (current 5 and 6), since they are explaining results from different samples, we think that it would be rather not be merged.

Finally, figure 8 and 9 have been merged (to current figure 7) following reviewer suggestion. 

7)      Fig. 5 is not mentioned in the text

A sentence has been added.

Reviewer 2 Report

Manuscript Number: applsci-2237059

This research has been conducted to characterize and validate the resistive heating as a synthesis method for zirconium oxides (ZrO2). A wire of Zr has been oxidized to form a core shell structure, in which the core is the metal wire, and the shell is an oxide layer around 10μm thick. Although this research is not very innovative, I think the data and results of this work can help researchers working in this field. There are few issues I would like to be taken in consideration, before any publication. Therefore, the reviewer suggests a minor revision of this work.

1.     In the abstract section, the sentences between lines 8 to 13 are repeated.

2.     It is necessary to mention innovation of the work at the end of the introduction section.

3.     English text is weak and should be edited. there are few typos and spaces missing. Some parts are not completely clear.

4.     It is necessary to rewrite the abstract by mentioning the important results of the work, quantitatively.

5.     It is necessary to perform EDS analysis of the surface of the wire. The results of the mentioned analysis are shown in Figure 6. The results of the mentioned analysis are shown in Figure 6

Author Response

Dear Editor

Concerning the manuscript referenced applsci-2237059, and entitled “Growth of Zr/ZrO2 core-shell structures by Fast Thermal Oxidation”, we have revised the manuscript according to the reviewers’ comments. We thank their carefully work  what, doubtlessly, will improve the quality of the paper. Find below the list of changes made in the manuscript, where they have been marked in yellow.

Reviewer #2

This research has been conducted to characterize and validate the resistive heating as a synthesis method for zirconium oxides (ZrO2). A wire of Zr has been oxidized to form a core shell structure, in which the core is the metal wire, and the shell is an oxide layer around 10μm thick. Although this research is not very innovative, I think the data and results of this work can help researchers working in this field. There are few issues I would like to be taken in consideration, before any publication. Therefore, the reviewer suggests a minor revision of this work. 

  1. In the abstract section, the sentences between lines 8 to 13 are repeated.

Abstract has been rewritten.

  1. It is necessary to mention innovation of the workat the end of the introduction section.

A paragraph has been added.

  1. English text is weak and should be edited. there are few typos and spaces missing. Some parts are not completely clear.

English has been revised by a native speaker.

  1. It is necessary to rewrite the abstract by mentioning the important results of the workquantitatively.

Abstract has been rewritten.

  1. It is necessary to perform EDS analysis of the surface of the wire. The results of the mentioned analysis are shown in Figure 6.The results of the mentioned analysis are shown in Figure 6

      We do not understand what the reviewer refers with this comment to figure 6. EDX spectrum on the surface of the wire is shown in figure 7. The figure caption has been rewritten to clarify this point.

Round 2

Reviewer 1 Report

Dear Authors,

I appreciate your explanations. But still, I have concerns about XRD data. Your responses showed a good-quality diffractogram, and I do not understand why you claim it is difficult to analyse. In my opinion, there is even a possibility for the calculation of lattice parameters for crystallographic phases which are presented in the sample.
Hence, I recommend placing XRD data within the article in the form, as you showed in your responses and doing basic crystallographic calculations.

Author Response

Dear Editor

Concerning the manuscript referenced applsci-2237059, and entitled “Growth of Zr/ZrO2 core-shell structures by Fast Thermal Oxidation”, we have revised the manuscript according to the reviewer’s comments. We thank his/her work. Find below the list of changes made in the manuscript, where they have been marked in yellow.

Reviewer #1

Dear Authors,

I appreciate your explanations. But still, I have concerns about XRD data. Your responses showed a good-quality diffractogram, and I do not understand why you claim it is difficult to analyse. In my opinion, there is even a possibility for the calculation of lattice parameters for crystallographic phases which are presented in the sample.

Hence, I recommend placing XRD data within the article in the form, as you showed in your responses and doing basic crystallographic calculations.

We have included XRD data as part a of figure 3 including the phase analysis. We have made calculations of the lattice parameters, but as the signal is poor, the accuracy of the calculated values is too low, so we think it is not worthy to include the values in the text.
